# Glutamine Produces Ammonium to Tune Lysosomal pH and Regulate Lysosomal Function

**DOI:** 10.3390/cells12010080

**Published:** 2022-12-24

**Authors:** Jian Xiong, Thi Thu Trang Luu, Kartik Venkatachalam, Guangwei Du, Michael X. Zhu

**Affiliations:** 1Department of Integrative Biology and Pharmacology, McGovern Medical School, The University of Texas Health Science Center at Houston, Houston, TX 77030, USA; 2Program in Biochemistry and Cell Biology, MD Anderson Cancer Center UTHealth Graduate School of Biomedical Sciences, Houston, TX 77030, USA; 3Program in Neuroscience, MD Anderson Cancer Center UTHealth Graduate School of Biomedical Sciences, Houston, TX 77030, USA

**Keywords:** amino acid starvation, autophagy, autophagosome, glutaminase, glutamine, lysosomal pH, mTORC1 activation

## Abstract

Glutamine is one of the most abundant amino acids in the cell. In mitochondria, glutaminases 1 and 2 (GLS1/2) hydrolyze glutamine to glutamate, which serves as the precursor of multiple metabolites. Here, we show that ammonium generated during GLS1/2-mediated glutaminolysis regulates lysosomal pH and in turn lysosomal degradation. In primary human skin fibroblasts BJ cells and mouse embryonic fibroblasts, deprivation of total amino acids for 1 h increased lysosomal degradation capacity as shown by the increased turnover of lipidated microtubule-associated proteins 1A/1B light chain 3B (LC3-II), several autophagic receptors, and endocytosed DQ-BSA. Removal of glutamine but not any other amino acids from the culture medium enhanced lysosomal degradation similarly as total amino acid starvation. The presence of glutamine in regular culture media increased lysosomal pH by >0.5 pH unit and the removal of glutamine caused lysosomal acidification. GLS1/2 knockdown, GLS1 antagonist, or ammonium scavengers reduced lysosomal pH in the presence of glutamine. The addition of glutamine or NH_4_Cl prevented the increase in lysosomal degradation and curtailed the extension of mTORC1 function during the early time period of amino acid starvation. Our findings suggest that glutamine tunes lysosomal pH by producing ammonium, which regulates lysosomal degradation to meet the demands of cellular activities. During the early stage of amino acid starvation, the glutamine-dependent mechanism allows more efficient use of internal reserves and endocytosed proteins to extend mTORC1 activation such that the normal anabolism is not easily interrupted by a brief disruption of the amino acid supply.

## 1. Introduction

Lysosomes play important roles in the degradation of proteins arriving from extracellular space and plasma membrane through endocytosis and from cytosol and intracellular organelles via three forms of autophagy, macroautophagy, chaperone-mediated autophagy, and microautophagy. The degradation function is carried out by the 60 or so hydrolases residing in the lysosome lumen. Interestingly, most of the hydrolases have optimal pH in the acidic range [1], which more or less matches the acidic environment of the lysosome lumen [2,3]. However, neither the optimal pH of all the hydrolases nor the luminal pH of the lysosomes is kept at a constant value, implicating that the lysosomal pH may be continuously tuned to meet the demand of degradation of different proteins under different conditions.

The acidic luminal pH of lysosomes is created by the vacuolar H^+^-ATPase (V-ATPase), which is abundantly present on the lysosomal membrane and transports protons against its gradient from the cytosol into the lysosome at the expense of hydrolyzing ATP [2]. The acidity of the lysosomal lumen is generally thought to be maintained at around pH 5, with differences considered representing distinguished lysosomal subpopulations carrying out specific functions or at different stages of their biosynthesis or function [3]. However, how lysosomes respond to the on-demand protein degradation needs such as those associated with autophagy or protein endocytosis remain completely mysterious.

Macroautophagy (herein referred to as autophagy) is a common phenomenon that occurs in all cells when nutrients, growth factors, amino acids, and glucose, become limited. It represents a drastic change in which the cell shuts down its synthetic pathways and devotes most of its resources to degradation so that sufficient building blocks, amino acids, sugars, and lipids, can be made from breaking down unwanted or nonessential cellular materials to sustain the synthesis of macromolecules essential for survival [4,5,6]. The mechanistic target of rapamycin (mTOR) is the master regulator of cell metabolism. When nutrients are abundant, mTOR complex 1 (mTORC1) promotes the synthesis of cellular materials and suppresses autophagy to a level that only allows clearance of unwanted intracellular components [7,8,9]. During nutrient shortage, or starvation, mTORC1 activity is reduced, which not only slows down anabolism but also lifts the inhibition on autophagy. Interestingly, growth factors, amino acids, and glucose exert effects on mTORC1 via distinct pathways, which converge at the lysosomal membrane to support mTORC1 activation [10]. For this reason, the deprivation of growth factors (by serum depletion), amino acids, or glucose may also affect mTORC1 and thereby autophagy differently in terms of kinetics and magnitudes. More importantly, in the absence of starvation, cells maintain a basal level of autophagy for quality control and selective degradation of wastes and labeled cargoes [11,12,13,14,15,16]. This basal degradation pathway is likely independent of mTORC1 regulation, but how it responds to starvation remains to be elucidated.

Recently, it was shown that deprivation of amino acids, but not serum, rapidly induced degradation of selective autophagy cargo proteins, such as p62/SQSTM1 (Sequestosome-1), TAX1BP1 (Tax1 Binding Protein 1), NDFIP1 (Nedd4 Family Interacting Protein 1), and CALCOCO2 (Calcium Binding and Coiled-Coil Domain 2, also known as NDP52 for nuclear dot protein of 52 kDa) in several human cell lines independently of mTORC1 and an increase in autophagosome synthesis, suggesting that amino acid availability can impact lysosomal protein degradation without involving mTORC1 [17]. Here, we identify glutamine as the key amino acid species that regulates lysosomal protein degradation. By producing ammonium through glutaminase-mediated hydrolysis, the millimolar concentration of glutamine commonly present in culture medium keeps lysosomal pH about 0.5–1 unit higher than pH 5.0 under resting conditions, resulting in a low degradation capacity. Either amino acid withdrawal or glutamine depletion causes the lysosomal pH to decrease, leading to accelerated degradation of LC3-II (lipidated microtubule-associated proteins 1A/1B light chain 3B), selective autophagy receptor proteins, and endocytosed proteins, which in turn extends mTORC1 activity for minutes in the early stage of amino acid starvation. These findings reveal a previously unappreciated role of glutamine in regulating lysosomal function through fine tuning lysosomal pH using a by-product, ammonium, from glutaminolysis.

## 2. Materials and Methods

### 2.1. General Reagents and Antibodies

N-acetylcysteine (NAC), brefeldin A (BFA), chloroquine (CQ), epidermal growth factor (EGF), L-glutamine solution (200 mM), insulin, l-ornithine-l-aspartate (LOLA), 4-phenylbutyric acid (4-PBA), and all amino acids were purchased from Sigma-Aldrich (St. Louis, MO, USA). Oxo-2-glutarate was purchased from APExBio Technology (Houston, TX, USA). BPTES, monensin, and nigericin were purchased from Cayman Chemical Company (Ann Arbor, MI, USA). Bovine serum albumin (BSA), DQ-BSA (Cat # D12051), tetramethylrhodamine-BSA (TMR-BSA, Cat # D1868), fluorescein isothiocyanate (FITC)-conjugated dextran 10,000 MW (Cat # D1820), Oregon Green 488-conjugated dextran 10,000 MW (Cat # D7170), glucosamine (GlcN), and methyl pyruvate (MP) were purchased from ThermoFisher Scientific (Waltham, MA, USA). Antibodies for phospho-S6K (T389) (Cat # 9206), S6K (Cat # 9202), myc (Cat # 2276), and p62 (Cat # 39749S) were purchased from Cell Signaling Technology (Danvers, MA, USA), for LC3 (Cat # 7543), NDFIP1 (Cat # HPA009682), TAX1BP1 (Cat # HPA024432), CALCOCO2 (Cat # HPA023195), and Flag (Cat # F1804) were from Sigma-Aldrich, for actin (Cat # 47778) was from Santa Cruz (Dallas, TX), for SNAT7 (Cat # HPA041777) and SNAT9 (Cat # PA5-60509) were form ThermoFisher Scientific, for GLS1 (Cat # 156876) and GLS2 (Cat # ab113509) were from abcam (Cambridge, UK). Secondary antibodies Dylight 800-conjugated goat anti-mouse IgG (Cat # SA5-10176) and Dylight 680-conjugated goat anti-rabbit IgG (Cat # 35568) were from ThermoFisher Scientific.

### 2.2. cDNA Constructs, Lentivirus Production, and Transduction

The expression construct for LAMP1-GFP was previously described [18], for FUGW-PK-hLC3∆G was a gift from Isei Tanida (Addgene plasmid # 61461), for Flag-SNAT9 was a gift from David Sabatini (Addgene plasmid #71858), and for SNAT7 was amplified from total RNA prepared from HEK-293 cells by RT-PCR using primers: GGCGAGCTAGCTCGAGATATGGCCCAGGTCAGCATCAACAATG and TCGCGGCCGCGGATCCTCATGCCAAGAGATCCACAAAGATGGC. The PCR product was subcloned into the pCDH-3Xmyc vector at the *Xho*I and *Bam*HI sites using the Infusion method (Takara Bio, Kusatsu, Japan). All cDNA inserts were validated by DNA sequencing. For small hairpin RNA (shRNA)-mediated knockdown, lentiviral plasmids containing shRNA for mouse SNAT7 (two constructs, targeting sequences CCCTTCCTGTTTCCATCTTTA and CAGTGTCATGTGAGTAGTGTA), SNAT9 (two constructs, targeting sequences GCCTTGTATCAAGACACTAAA and CCTGGCTTTCGTGTTCATATA), and an shRNA with random sequence (CCTAAGGTTAAGTCGCCCTCG, referred to as shCtrl) in the pLOK.1 vector were purchased from Sigma-Aldrich. shRNAs for mouse GLS1 (targeting sequence AGAAAGTGGAGATCGAAATTT) and mouse GLS2 (targeting sequence CTCCCTCAATGAGGAAGGAAT) were purchased from VectorBuilder (Shenandoah, TX, USA).

For lentivirus production and transduction, TLA-293T cells (ThermoFisher Scientific) were co-transfected with the desired lentiviral vector (LAMP1-GFP, myc-SNAT7, Flag-SNAT9, shGLS1, shGLS2, shSNAT7, shSNAT9, or FUGW-PK-hLC3∆G), pCMV-dR8.2, and pMD2.G using Lipofectamine 3000 and Plus reagent (ThermoFisher Scientific) as previously described [18]. The virus-containing culture media collected at 28 and 52 h post transfection were pooled. For infection of mouse embryonic fibroblast (MEF) and L929 cells, polybrene was added to the collected medium at 8 µg/mL before application to the cells at the volume equal to that of the culture medium. After an overnight incubation, viruses were removed, and cells were incubated in fresh medium for 2 days. On day 3 after the lentivirus infection, 1 µg/mL puromycin was added to the medium for selection of cells with stable expression. Cells were used for experiments within 1 to 2 weeks.

### 2.3. Cell Lines and Culture Media

MEF cells were prepared as described [19]. Briefly, pregnant mice were sacrificed at 10–13 days post-coitum, and embryos were collected from uterine horns. After removing head and red organs, the remaining embryos were washed, finely minced, and dissociated with trypsin. After centrifugation and culture in gelatin-coated flasks, cells were frozen for future use. All other cells were purchased from American Type Culture Collection (Manassas, VA, USA) unless indicated otherwise.

All cells were maintained in Dulbeco’s modified Eagle’s medium (DMEM, high glucose, GenDepot, Katy, TX, USA) supplemented with 10% fetal bovine serum (FBS, GenDepot), 100 unit/mL penicillin and 100 µg/mL streptomycin (GenDepot) at 37 °C, 5% CO_2_. For glutamine deprivation and amino acid starvation, cells were treated with glutamine-free DMEM (Sigma, Cat # 5671) and amino acid-free DMEM (United State Biological, Salem, MA, USA), respectively, supplemented with 10% FBS that was pre-dialyzed with Slide-A-Lyzer™ Dialysis Cassettes (ThermoFisher Scientific, Cat # 66110) to remove molecules smaller than 3.5 kDa. DMEM omitting individual amino acids were prepared from the amino acid-free DMEM with the addition of desired combinations of amino acids diluted from the following stock solutions: L-glycine (400 mM, 1000×), L-glutamine (1 M, 250×), L-arginine (400 mM, 1000×), L-cysteine (200 mM, 1000×), L-histidine (200 mM, 1000×), L-isoleucine (800 mM, 1000×), L-leucine (800 mM, 1000×), L-lysine (800 mM, 1000×), L-methionine (200 mM, 1000×), L-phenylalanine (400 mM, 1000×), L-serine (400 mM, 1000×), L-threonine (800 mM, 1000×), L-tryptophan (40 mM, 1000×), L-tyrosine (400 mM, 1000×), and L-valine (800 mM, 1000×).

### 2.4. Western Blot Analysis

Cells were seeded in 6-well plates and grown overnight. At the time of the experiment, the cell density was 60~80%. On the day of the experiment, the old culture medium was replaced with fresh medium at 1 h before the treatment. After the treatment, cells were immediately lysed in the well by the addition of 150 µL 1 × SDS sample buffer (50 mM Tris, 12.5 mM EDTA, 2% SDS, 10% glycerol, 2% β-mercaptoethanol, 0.02% bromophenol blue, pH 6.8). Lysates were collected in a 1.6-mL microcentrifuge tube and briefly sonicated before incubation at 95 °C in a heat block for 10 min. After cooling, the lysates were loaded on SDS-PAGE (7.5% acrylamide for all proteins except for LC3 and actin which used 15% acrylamide) and transferred onto nitrocellulose membranes (Li-COR Biotechnology, Lincoln, NE, USA) for all proteins except for LC3 and actin which used low fluorescence background PVDF membranes (MilliporeSigma, Burlington, MA, USA). The membranes were blocked in 5% non-fat milk for 1 h at the room temperature (21–23 °C) and probed with the indicated primary antibodies overnight at 4 °C. Dylight 680 and Dylight 800 conjugated secondary antibodies were used for fluorescently labeling primary antibodies and detection by the Li-COR Odyssey infrared imaging system (Li-COR). Quantification of band intensity was performed from at least 3 repeats with Li-COR ImageStudio Lite software.

### 2.5. Ratiometric Lysosome pH Measurement 

Lysosomal pH measurement was performed as previously described [20]. Cells were seeded at a density of 30~50% on glass bottom dishes (MatTek, Ashland, MA, USA) one day before imaging. Oregon Green 488- or FITC-conjugated dextran was added to cells at the final concentration of 0.2 mg/mL during seeding. After overnight incubation of the dextran, cells were washed 3 times with DMEM containing 10% FBS and chased with fresh medium for 2 h. After the chase, the medium was replaced with phenol red-free DMEM containing the desired concentration of glutamine supplemented with 10% pre-dialyzed FBS. Cells were incubated for 1 h (unless otherwise indicated) before imaging in the same medium used for the treatment. During imaging, cells were placed in an environmental chamber with temperature (37 °C) and CO_2_ (5%) control (Tokai Hit USA Inc., Bala Cynwyd, PA, USA). Images were acquired using Nikon NIS Elements software on a Nikon A1R confocal microscope equipped with 40× oil immersion lens. Excitation wavelengths were switched between 445 nm and 488 nm lasers. Emissions were taken with an EGFP filter. Each individual lysosome or a cluster of lysosomes was selected as one ROI (region of interest), for which the mean intensity ratio (488/445) was calculated. pH calibration was performed in isotonic K^+^ solutions (140 mM KCl, 1 mM MgCl_2_, 1 mM CaCl_2_, 5 mM glucose, supplemented with 10 μM nigericin), with pH of 4.0, 4.5, 5.0, 5.5, 6.0, 6.5, and 7.0, respectively, applied sequentially to cells. For each pH, images were taken after a 10 min incubation with the calibration buffer in the environmental chamber at 37 °C, 5% CO_2_. The mean values of the 480/445 ratios of the individual ROIs were fitted as a function of pH to a Boltzmann sigmoidal curve, and the lysosome pH values were then calculated based on the equation obtained from the curve. The frequency distribution of lysosomal pH values was analyzed using Prism 7 (GraphPad Software, Inc., San Diego, CA, USA) and fitted into Gaussian distribution.

For time lapse imaging of glutamine-induced lysosomal pH changes, after overnight FITC-dextran loading, L929 cells were chased in normal culture medium for 2 h. This was followed by incubation with 4-BPA (10 mM), LOLA (10 mM), BPTES (20 µM), or vehicle in the normal culture medium for 1 h and subsequently in the glutamine-free medium for another hour. Fluorescence imaging began immediately upon addition of 2 mM glutamine to the medium with the 488/445 ratio acquired at 10-s intervals. For the drug treated cells, 4-PBA, LOLA, and BPTES were continuously present throughout.

For lysosomal pH monitoring using pHluorin (Green) and mKate2 (red) ratios, MEF cells infected with lentivirus for PK-hLC3 were placed in the glutamine-free medium supplemented with 10% pre-dialyzed FBS for 1 h before images for both green (ex/em: 488/525 nm) and red (ex/em: 560/595 nm) fluorescence were taken at 37 °C, 5% CO_2_. The same cell was imaged again 1 min after the addition of 4 mM glutamine.

### 2.6. DQ-BSA Degradation Assay

Cells were seeded at a density of 30~50% on glass bottom dishes one day before imaging. On the next day, the medium was replaced with fresh medium containing 0.1 mg/mL DQ-BSA and incubation continued for 2 h. After the incubation, the cells were washed three times with the culture medium and chased for 1 h. After the chase, the medium was replaced with fresh complete DMEM (control), amino acid-free DMEM, or glutamine-free DMEM supplemented with 10% pre-dialyzed FBS and cells incubated for 1 h before being imaged in an environmental chamber at 37 °C, 5% CO_2_. Z-stack images were taken using a Nikon A1R confocal microscope with 60× oil immersion lens. Quantification analysis was performed using ImageJ by measuring the average fluorescence intensities of individual cells.

### 2.7. BSA Endocytosis Assay 

For DQ-BSA uptake assay, MEF cells stably expressing LAMP1-GFP were seeded on glass bottom dishes at confluence of 30–40%. After overnight culture, the medium was replaced with fresh culture medium and the culture continued for 1 h. Cells were then incubated with DQ-BSA (0.1 mg/mL) in normal DMEM (4 mM glutamine) or glutamine-free DMEM (both phenol red-free) supplemented with 10% pre-dialyzed FBS for 1 h. Images were taken with a Nikon A1R confocal microscope at 37 °C, 5% CO_2_, for both green (ex/em: 488/525 nm) and red (ex/em: 560/595 nm) fluorescence. For TMR-BSA uptake assay shown in Appendix A, MEF cells stably expressing LAMP1-GFP were processed as above except that TMR-BSA was used in place of DQ-BSA and the incubation time was 2 h instead of 1 h. After the 2-h TMR-BSA incubation, cells were immediately washed twice in the same medium used for incubation before images were taken within 10 min after washing. Pearson correlation coefficients analysis was performed using the Coloc 2 plugin of ImageJ.

### 2.8. Data Analysis

All statistical analyses were performed using Prism 7. Paired or unpaired two-tail Student’s *t* test or ANOVA were used as appropriate, with *p* < 0.05 considered statistically significant. Summary data show mean ± SEM unless indicated otherwise.

## 3. Results

### 3.1. Amino Acid Starvation Accelerates Lysosomal Degradation

Starvation leads to mTORC1 inactivation and autophagy upregulation. As a marker of autophagosomes, the level of LC3-II is usually increased in response to the withdrawal of either both serum and amino acids or just amino acids from the culture media. However, during the early hours of starvation, LC3-II could also exhibit a decrease [21,22,23]. Indeed, in primary mouse embryonic fibroblast (MEF) cells exposed for 1 h to Hank’s Buffered Salt Solution (HBSS) or amino acid-free Dulbecco’s Modified Eagle Medium (DMEM) containing 10% fetal bovine serum (FBS), LC3-II levels were decreased by ~50%, expressed in both LC3-II/LC3-I ratios (Figure 1A,B) and LC3-II levels normalized to the loading control, actin (Appendix A). No significant change was seen with LC3-I (Appendix A). The LC3-II decrease was not detected in cells exposed to serum-free DMEM that contained normal amino acid content for 1 h (Figure 1A,B, Appendix A), suggesting it to be caused mainly by amino acid starvation.

With the lysosomal degradation inhibited by chloroquine (CQ, 100 µM), the starvation-induced decrease in LC3-II was prevented (Figure 1A,B; Appendix A), indicating that enhanced lysosomal degradation underpins the decrease in LC3-II induced by amino acid starvation. CQ also increased LC3-II levels under control conditions, suggesting that there is basal turnover of autophagy in well-fed cells (Figure 1B, Appendix A). The marked decrease in the level of phospho-p70S6K (pS6K), a marker for mTORC1 activity, in the starved cells confirmed that macroautophagy was triggered under all starvation conditions (Figure 1A).

In addition to the primary MEF cells, the short-term amino acid starvation also reduced LC3-II levels in two immortal cell lines, HeLa (human epithelial cells) and L929 (mouse fibroblasts), which was blocked by CQ (Appendix A), suggesting this being a common mechanism. In human primary skin fibroblasts, BJ cells, we observed similar LC3-II reduction upon 1-h amino acid withdrawal (Figure 1C). Moreover, multiple selective autophagy receptor proteins, such as p62 (SQSTM1, for sequestosome 1), NDFIP1, TAX1BP1, and CALCOCO2 (NDP52), were also reduced in the starved cells (Figure 1C). Previously, these selective autophagy receptors were reported to be decreased after 1-to-4-h starvation and taken as evidence of endosomal microautophagy that occurs independently of mTORC1 inactivation [17].

To learn if the enhanced lysosomal degradation is specific for autophagosome cargoes or general to all forms of degradation that occur in the lysosome, we used DQ-Red BSA (DQ-BSA), a fluorogenic substrate essentially made of bovine serum albumin (BSA) conjugated with self-quenched BODIPY dyes. DQ-BSA is taken up by the cell through endocytosis and then trafficked to lysosomes for degradation. The resultant dequenching of the fluorophores leads to bright fluorescence [24]. To deliver the substrate to lysosomes through endocytosis, we pulse labeled MEF cells with DQ-BSA for 2 h and then chased for 1 h. Cells were then cultured in the regular culture medium or amino acid-free medium for 1 h before images were taken. As shown in Figure 1D,E (also see Appendix A), the starved MEF cells exhibited about twice the fluorescence intensity compared to the control cells, suggesting that the degradation of DQ-BSA was enhanced upon amino acid withdrawal from the medium. Given that DQ-BSA was preloaded before amino acid withdrawal, this increase in fluorescence represented accelerated degradation of DQ-BSA that already existed in lysosomes in response to amino acid starvation. Together, these data demonstrate that short-term amino acid starvation triggers a general increase in lysosome degradation capability.

### 3.2. Glutamine Starvation Alone Increases Lysosomal Degradation

The DMEM culture medium contains 15 different amino acid species, with a total concentration of 10 mM. To know if the increased lysosomal degradation was caused by the shortage of a specific amino acid species or the general loss of total amino acids, we formulated 15 amino acid-depleted media (DMEM) omitting the individual amino acid species one at a time. Interestingly, by removing only glutamine (Gln) for 1 h, we detected ~50% decrease in LC3-II in MEF cells, while the exclusion of any other amino acid species did not significantly affect the LC3-II levels (Figure 2A; Appendix A).

Since Gln is the most abundant amino acid species (4 mM) in DMEM, we asked whether the decreased LC3-II might result from a nonspecific effect due to the large decrease in total amino acid concentration upon omitting Gln. To test this possibility and learn whether the effect of Gln could be fulfilled by another amino acid, we substituted Gln in DMEM with the equal concentration of another amino acid individually and tested the effect of 1-h treatment on LC3-II levels in MEF cells. However, none of the amino acid substitutions rescued the decreased LC3-II level found in the Gln-deprived DMEM, including asparagine (Asn) and glutamate (Glu) (Figure 2B; Appendix A), which are structurally the most similar to Gln. In the DQ-BSA assay, the starvation of Gln only for 1 h also increased the fluorescence by ~1-fold (Figure 2C,D, Appendix A), similar to the increase induced by total amino acid starvation (Figure 1D,E). These data further strengthen the unique role of Gln in regulating lysosomal degradation, showing that Gln shortage alone is sufficient to accelerate lysosomal degradation.

With MEF cells exposed to DMEM containing 0–10 mM Gln for 1 h, we observed a clear Gln concentration-dependent increase in LC3-II levels beginning at 2 mM Gln and the effect approached saturation at >8 mM Gln (Figure 2E,F; Appendix A). By contrast, Gln did not change LC3-I levels (Appendix A). Importantly, while the normal DMEM contains 4 mM Gln, plasma Gln levels range from 200 µM–4 mM [25,26], indicating that most body cells are exposed to Gln concentrations that allow some levels of LC3-II accumulation. Furthermore, the regulation of LC3-II levels by Gln was independent of mTORC1 activity, as the pS6K levels remained unchanged in cells treated for 1 h with different Gln concentrations (Figure 2E). This result is consistent with the previous study showing that Gln starvation decreased LC3-II independently of mTORC1 [27]. Moreover, the withdrawal of Gln for 1 h also decreased LC3-II levels in HeLa and L929 cells without affecting mTORC1 activation (Appendix A).

To test whether selective autophagy receptor proteins also undergo degradation in response to Gln starvation, we subjected BJ cells to Gln deprivation from 20 min to 4 h. At 20 min after Gln withdrawal, while no significant reduction was detected for p62 and NDFIP1, LC3-II, TAX1BP1 and CALCOCO2 showed 50%, 30%, and 30% reductions, respectively. At 40 min, LC3-II, TAX1BP1 and CALCOCO2 were further decreased by 70%, 50%, and 40%, respectively, and p62 and NDFIP1 also showed reductions of 30% and 40%, respectively. The reductions approached the maximum for LC3-II, TAX1BP1, CALCOCO2, and p62 at 1 h, but for NDFIP1, it did not saturate until after 2 h of the Gln withdrawal (Figure 2G,H). Thus, Gln starvation represents a common mechanism of accelerating lysosomal degradation of autophagosome cargoes and autophagy receptors, although the rates of degradation may differ for different substrates.

### 3.3. Glutamine Regulates Lysosomal Degradation Not through Producing Glutathione and α-Ketoglutarate or Sensing by Known Lysosomal Glutamine Transporters

In addition to being a building block of protein synthesis, Gln also serves two major metabolic roles: (1) as a precursor of glutathione, the major reducing agent in the cells, and (2) as a precursor of α-ketoglutarate, an intermediate in the TCA cycle. Both functions require the conversion of Gln to glutamate (Glu). That supplementing Glu failed to rescue LC3-II to the control level in cells cultured in the Gln-free medium (Figure 2B) would suggest either that oxidation and/or α-ketoglutarate depletion were not the reasons for the altered lysosomal degradation, or that the uptake and/or utilization of Glu by the cell were not as efficient as that of Gln, since different amino acid transporters are used for Glu and Gln [28]. To distinguish these possibilities, we inhibited oxidation by using a membrane permeable reducing agent, N-acetylcysteine (NAC, 6 mM), and supplemented the cells with a membrane permeable analog of α-ketoglutarate, dimethyl-2-oxoglutarate (DM-2-OG or simply D2O, 6 mM), respectively, aiming to see if either or both of these could negate the need for Gln to maintain the normal lysosomal degradation capability. However, neither NAC nor DM-2-OG, nor the combination of both, prevented the LC3-II decrease caused by Gln withdrawal (Figure 3A,B; Appendix A). The treatment with DM-2-OG even led to ~40% decrease in LC3-II/LC3-I ratio in the normal medium that contains Gln (Figure 3A,B). Likewise, the accelerated degradation of selective autophagy receptors, p62, NDFIP1, TAX1BP1 and CALCOCO2, caused by Gln starvation was not altered by the inclusion of Glu, DM-2-OG or NAC either (Appendix A). Therefore, it is unlikely that Gln deprivation accelerated lysosomal degradation due to decreases in the production of glutathione and/or α-ketoglutarate.

Next, we asked if the observed Gln regulation of lysosomal degradation represents a sensing mechanism to Gln availability. Gln has been suggested to be sensed by Arf1 (ADP-ribosylation factor 1) to regulate mTORC1 activation, and inhibition of Arf1 by brefeldin A (BFA) abolished this effect [29]. However, a treatment of BFA (10 µM) reduced pS6K levels without affecting the ability of Gln withdrawal to decrease the LC3-II levels (Figure 3C). Moreover, it was shown that the Na^+^-dependent lysosomal amino acid transporter, SNAT9 (also known as SLC38A9), also functions as a sensor that conveys the information about amino acid abundance in the lysosomal lumen to mTORC1, and overexpressing SNAT9 enables the cell to bypass the luminal amino acid requirement, causing persistent mTORC1 activation despite the amino acid shortage [30,31,32]. SNAT7 (or SLC38A7) is another transporter belonging to the same family of SNAT9 that conducts Gln transport across the lysosomal membrane [33,34]. To examine whether these lysosomal amino acid transporters are involved in regulating lysosomal degradation in response to Gln withdrawal, we altered the expression of SNAT7 or SNAT9 in MEF cells. However, Gln withdrawal still reduced LC3-II levels in cells that overexpressed Myc-SNAT7 or Flag-SNAT9 (Figure 3D,E) as well as in cells that had SNAT7 or SNAT9 knocked down by short-hairpin (sh) RNA (Figure 3F,G), similarly as the parental MEF cells and those transfected with the control shRNA. Taken together, these results indicate that the currently known lysosomal amino acid/Gln sensing mechanisms are not responsible for the rapid decrease in LC3-II levels caused by extracellular Gln depletion.

### 3.4. Glutamine Influences Lysosomal pH by Producing Ammonia in a Glutaminase-Dependent Manner

Aside from producing glutathione and α-ketoglutarate, the same reaction that hydrolyzes Gln to Glu also yields ammonium. Since ammonium can exist in both charged (NH_4_^+^) and uncharged (NH_3_) forms, it is not only freely diffusible across cellular membranes, including lysosomal membranes, but also able to impact pH inside the organellar lumen. The more acidic pH inside the lysosome than in the cytosol means that when NH_3_ enters the lysosome, a higher proportion of it will be bound by protons than when it is in the cytosol. This will result in a loss of free protons, i.e., an increase in the lysosomal luminal pH [35]. To test this possibility, we measured lysosomal luminal pH using dextran-conjugated Oregon Green 488 (OG-488), a ratiometric pH indicator taken up to lysosomes through endocytosis [36]. With MEF cells loaded with OG-488 and maintained in FBS-containing DMEM with 0 or 6 mM Gln at 37 °C for 1 h, the median pH value of individual lysosomes shifted from 4.86 in the Gln-free medium to 5.50 in 6 mM Gln (Figure 4A,B). Pseudocolor images of lysosomal pH revealed that in medium with either 0 or 6 mM Gln, lysosomes near the nuclei displayed lower pH than those in the cell peripheral (Figure 4A, enlarged images), consistent with the previous report [3]. Moreover, the span of the pH range remained approximately the same, showing a parallel shift of the frequency distribution curve to higher pH in the presence of Gln (Figure 4C). This indicates that, rather than targeting a certain population(s) of lysosomes, Gln elevates lysosomal pH uniformly in the cell, consistent with the freely diffusible nature of ammonia.

Next, using pHluorin-mKate2-hLC3 (PK-hLC3) as a genetically encoded probe [37] to indicate luminal pH of autolysosomes, we observed mostly orange puncta of LC3 in MEF cells exposed to the Gln-free culture medium (Appendix A). This is consistent with a stronger quenching of pHluorin (green) than mKate2 (red) by the acidic pH in the lysosomal lumen. However, within 1 min of addition of Gln (4 mM) back to the culture medium, the fluorescence intensity increased in both channels, with the increase in the green channel being more pronounced than that in the red channel such that the LC3 puncta became bright yellow (Appendix A), suggesting that Gln causes a rapid increase in lysosomal pH.

To determine the upper limit of the Gln effect on lysosomal pH, we used FITC-dextran because of its higher pKa (~5.9) [38] and higher fluorescence ratio than OG-488 (Appendix A). We also used L929 cells for ease of gene manipulation. Similar to MEF cells, the median value of lysosomal pH in L929 cells increased from 4.69 in the Gln-free medium to 5.41 and 5.74 in the media that contained 2 and 4 mM Gln, respectively (Appendix A). That 10 mM Gln did not further alkalinize the lysosome than 4 mM Gln (Appendix A, median pH = 5.74) suggests that the ability of extracellular Gln to raise lysosomal pH saturates at ~4 mM in these cells.

Gln is hydrolyzed to Glu and ammonium by glutaminases (GLS1 and GLS2). To test if the glutaminases are involved in Gln regulation of lysosomal luminal pH, we knocked down GLS1 and GLS2 in L929 cells (Appendix A). While in 4 mM Gln, the median pH value decreased 0.32 pH unit, from 5.67 to 5.35 in the GLS knockdown (KD) cells (Figure 4D), with the distribution curve shifted to lower pH ranges (Figure 4E), in the Gln-free medium, the median values (pH 4.69 for shCtrl, pH 4.67 for shGLS1/2) and distribution of lysosomal pH were not affected (Figure 4F,G). In 2 mM Gln, GLS KD also decreased the median lysosomal pH by 0.32 unit from 5.41 to 5.09 (Appendix A); however, in 10 mM Gln, the change was only 0.17 pH unit (5.70 to 5.53) (Appendix A). Both the saturation effect of >4 mM Gln on lysosomal pH and the incomplete loss of the GLS activity may account for this effect, as the residual GLS in the KD cells could still produce ammonia, the level of which would likely be higher at 10 mM than at 2 and 4 mM Gln.

Moreover, with treatment of a selective GLS1 inhibitor, BPTES (20 µM, 1 h), the lysosomal pH of L929 cells grown in 4 mM Gln also markedly shifted to lower values, with the median pH of 5.52, although the decrease was not as pronounced as that seen in the GLS KD cells (Figure 4D,E). Importantly, BPTES did not further decrease lysosomal pH in the GLS KD cells (Figure 4D,E). In 2 mM Gln, the effect of BPTES was comparable to, or slightly better than, that caused by GLS KD, with the median pH of 5.02 (Appendix A). Since BPTES exhibits a much higher affinity to GLS1 than GLS2 [39], it is plausible that GLS1 is the major enzyme that produces ammonium from Gln in these cells.

To determine how quickly extracellular Gln alkalizes the lysosome, we performed time-lapse imaging using L929 cells loaded with FITC-dextran. We first incubated L929 cells in the Gln-free medium for 2 h to bring down the lysosomal pH to ~4.7 and then added 2 mM Gln to the medium while alternately acquiring FITC fluorescence images at 445 and 488 nm excitation at 10-s intervals. As shown in Figure 5A (black), on average, the luminal pH of the lysosomes increased to 5.8 within 90 s and then slowly declined until reaching a steady state at about 240 s. However, in the presence of 20 µM BPTES (added at 2 h before Gln and maintained throughout), the 2 mM Gln-induced increase was curtailed after the first 30 s and continued to rise only at a very slow rate (Figure 5A, blue). These results, demonstrating that inhibition of GLS1 indeed dramatically slows down Gln-evoked lysosomal alkalization, support the role of GLS1 and therefore glutaminolysis in Gln regulation of lysosomal luminal pH. Furthermore, BPTES also reduced NDFIP1 and TAX1BP1 levels in BJ and MEF cells (Figure 5B,C), indicating that inhibiting GLS1 accelerates the breakdown of these autophagy receptors.

To further probe if ammonium is indeed the product from Gln hydrolysis that is responsible for the alkalizing effect of Gln, we used ammonium scavengers, 4-phenylbutyric acid (4-PBA, 10 mM) and L-ornithine-l-aspartate (LOLA, 10 mM) [40,41]. Both 4-PBA and LOLA attenuated the Gln-evoked increase in lysosomal pH similarly as inhibiting GLS1 (Figure 5A, green for LOLA and red for 4-PBA). Moreover, as a membrane permeable agent that scavenges cytosolic ammonium through transamination of pyruvate to alanine [42], methyl pyruvate (MP, 20 mM) also lowered median lysosomal pH of MEF cells cultured in 4 mM Gln from 5.54 to 5.32 (Figure 5D,E). Conversely, the amino sugar glucosamine (GluN, 10 mM), which can enter glycolysis following deamination of glucosamine-6-phosphate, a reaction that produces ammonium [43], increased the median lysosomal pH to 5.78 (Figure 5D,E). Consistent with the changes in lysosomal pH, the basal LC3-II levels in MEF cells grown in normal culture medium containing 4 mM Gln were increased by the treatment of GluN (10 mM, 1 h) but decreased by that of MP (20 mM, 1 h) (Figure 5F). Since α-ketoglutarate can also act as an ammonium scavenger through the reverse reaction of glutamate dehydrogenase that makes Glu from α-ketoglutarate and NH_4_^+^, these findings can also explain the effect of D2O on lowering LC3-II levels in the normal medium (Figure 3A,B). Together, these data provide a strong support to the notion that cytosolic production of ammonium strongly impacts lysosomal pH regulation and in turn protein degradation. Moreover, Gln is the main source of the ammonium generation due to GLS1-mediated glutaminolysis.

### 3.5. Glutamine Sensing Confers Sustained mTORC1 Activity during Early Time Periods of Amino Acid Shortage

The above data suggest that under normal fed conditions, Gln through hydrolysis by GLS produces ammonium to raise the lysosomal luminal acidity to approximately 0.6–1.0 pH unit higher than that would be attained by V-ATPase. This keeps the lysosomal hydrolases at suboptimal conditions, matching the relatively slow rate of autophagic flux in resting cells. Upon starvation, the loss of Gln supply diminishes ammonium production so that the lysosomal luminal pH returns to the level set by the V-ATPase, which in turn boosts the hydrolase activities to accelerate lysosome degradation. Thus, the increased lysosomal degradation soon after starvation is simply driven by Gln depletion, rather than a response to the increased demand from macroautophagy for breaking down more materials. This implies that under conditions of amino acid shortage, the increase in lysosomal degradation may precede the activation of macroautophagy, a process driven by mTORC1 inactivation. To test this possibility, we measured LC3-II and pS6K levels in MEF cells at different times after amino acid withdrawal. Indeed, while LC3-II levels began to decrease at 5 min, the earliest time point measured, after the amino acid withdrawal, and continued to drop until 30 min to ~50% of the original value, the pS6K levels were maintained during the first 15 min and then dropped abruptly in the next 5 min, which was then followed by a slow and gradual declined to ~35% of the beginning level at 1 h (Figure 6A–C). These results indicate that increased lysosomal degradation is a more immediate response to amino acid starvation than mTORC1 inactivation.

To evaluate the role of Gln in starvation-induced lysosomal degradation and mTORC1 inactivation, we included 10 mM Gln in the amino acid-free medium and performed the same time course measurement as above. As expected, in the presence of Gln but absence of all other amino acids, LC3-II levels did not decrease over the course of 1 h. Rather, there was a slow and steady increase in the LC3-II levels to ~40% above the basal level after 1 h (Figure 6A,B; Appendix A), probably due to increased autophagosome synthesis. The addition of NH_4_Cl (10 mM) during amino acid starvation also prevented LC3-II decrease (Figure 6A,B; Appendix A). Additionally, the inclusion of Gln or CQ, but not Glu, in the amino acid-free medium reversed the reduction in LC3-II and selective autophagy receptor proteins, p62, NDFIP1, TAX1BP1, and CALCOCO2, in BJ cells (Appendix A). These results are consistent with the idea that Gln is the key factor for the acceleration of lysosomal degradation caused by amino acid starvation. Interestingly, both Gln and NH_4_Cl accelerated mTORC1 inactivation in the amino acid-starved cells, in which pS6K levels decreased by 50.7 ± 5.1% (*n* = 6, Gln) and 44.8 ± 8.9% (*n* = 6, NH_4_Cl) at 15 min of the starvation, as opposed to the 6.8 ± 5.0% (*n* = 6) decrease under control conditions, i.e., amino acid starvation in the absence of added Gln or NH_4_Cl (Figure 6A,C). These data suggest that the accelerated lysosome degradation during the early phase of amino acid starvation serves a special function in sustaining the mTORC1 activity, and the shortage of Gln is particularly critical for the increased lysosomal degradation that supports this grace period before mTORC1 inactivation.

### 3.6. Glutamine Deprivation Enhances Protein Endocytosis to Sustain mTORC1 Activity during Early Time Periods of Amino Acid Shortage

The removal of Gln has been shown to upregulate macropinocytosis [44,45,46]. This means that in response to Gln depletion, extracellular proteins, such as albumin, a major component of FBS, may be taken up by the cells at an accelerated rate, transported along the endocytic pathway to lysosomes, and then degraded to produce amino acids to sustain mTORC1 activity. To test if protein endocytosis is increased in response to Gln deprivation, we first examined the uptake of tetramethylrhodamine-conjugated BSA (TMR-BSA) by MEF cells that expressed LAMP1-GFP in culture medium containing 4 mM Gln or no Gln. After a 2-h incubation with TMR-BSA (0.1 mg/mL), the cells were washed immediately in the same medium used for the incubation and imaged by confocal microscopy (Appendix A). Based on Pearson correlation coefficients, TMR-BSA was better localized with LAMP1-GFP in the absence of Gln than in its presence (Appendix A), supporting the idea that Gln depletion accelerates protein endocytosis.

The somewhat weaker TMR fluorescence in Gln-free than in 4 mM Gln medium, perhaps due to stronger quenching by the lower luminal pH in the endosomes and lysosomes as well as increased lysosomal degradation of the endocytosed TMR-BSA in the Gln-free medium, made it difficult to see the enhanced endocytosis from the confocal images. Therefore, we repeated the experiments using DQ-BSA. DQ-BSA exhibits weak fluorescence signals before reaching the lysosome. Once reaching the lysosome, DQ-BSA is digested, giving rise to bright and discrete fluorescence signals that colocalized with LAMP1 (Figure 7A). In cells cultured in the Gln-free medium, DQ-BSA mainly exhibited well-defined discrete signals that colocalized with LAMP1-GFP (Figure 7A,B). However, in cells cultured in control medium containing 4 mM Gln, DQ-BSA showed considerably weaker fluorescence, which was not well colocalized with LAMP1-GFP (Figure 7A,B). Together with the data on TMR-BSA, these results demonstrate that Gln depletion enhances endocytic trafficking and subsequent digestion of endocytosed proteins in lysosomes.

We then asked whether the increased degradation of the endocytosed proteins contributes to the extension of mTORC1 activity during the early time periods of amino acid starvation. To allow a better control of extracellular proteins, we substituted FBS with 10 µg/mL insulin and 1 µg/mL epidermal growth factor (EGF) in the culture medium to provide the growth factors needed to support mTORC1 activity. When needed, 0.2% BSA was added to the medium to provide proteins for endocytosis. Cells were acclimated in the “serum-free” media for 1 h before amino acid starvation. Interestingly, in the presence of 0.2% BSA, pS6K levels remained unchanged for at least 15 min in the beginning of the amino acid withdrawal, and then declined gradually, reaching 94 ± 11% (*n* = 5), 67 ± 11% (*n* = 5) and 45 ± 10% (*n* = 5) of the basal level at 20, 30 and 60 min, respectively (Figure 7C,D). This was similar or slightly better than using the whole serum in terms of delaying mTORC1 inactivation during amino acid shortage (compared to Figure 6C). In the absence of BSA, however, pS6K decreased to 56.7 ± 3.3% (*n* = 3), 50 ± 0% (*n* = 3), and 43.3 ± 6.7% (*n* = 3) of the basal level at 20, 30, and 60 min, respectively, of the amino acid withdrawal (Figure 7C,D). These data suggest that in response to amino acid deprivation, cells take up more extracellular proteins by endocytosis as a source of amino acids to sustain mTORC1 activity.

To examine if lysosomal degradation is required for the protein uptake supported effect on mTORC1, we included Gln (10 mM) during amino acid starvation. Not only did Gln abolish the mTORC1 activity-extending effect of BSA, but it also accelerated the amino acid starvation-induced mTORC1 inactivation in the absence of BSA, albeit to a lesser extent than in the presence of BSA (Figure 7C,D). These results suggest that while lysosomal degradation of the endocytosed external proteins constitutes a major source of amino acids for sustaining the mTORC1 activity, the degradation of internal proteins arising from basal autophagy also plays a role. Taken together, our data demonstrate that lysosomal acquisition of extracellular proteins (e.g., BSA) is increased during amino acid starvation, and the endocytosed proteins is broken down by lysosomes together with the autophagic cargoes at an accelerated rate to produce new amino acids to sustain mTORC1 activity.

## 4. Discussion

### 4.1. Glutamine Regulates Lysosomal pH by Producing Ammonia from Glutaminolysis

Glutamine is one of the most abundant amino acids in the serum as well as culture medium. Aside from being a building block for protein synthesis, a precursor for cellular glutathione [47], a precursor of α-ketoglutarate that enters the TCA cycle for ATP synthesis [48], and a substrate for plasma membrane amino acid transporter SLC7A5 to aid the uptake of other amino acids [49], Gln has also been recurrently implicated in autophagy regulation, but with controversy. In several studies, Gln depletion in the culture media reduced LC3-II levels in various cell types [27,50,51]. Conversely, transcriptionally upregulating Gln synthetase, which converts Glu to Gln, enhanced LC3-II levels in a modified colon carcinoma cell line DLD1 that allowed for controlled activation of FOXO3 [52]. However, the same manipulation also inhibited lysosomal translocation and activation of mTORC1, which contrasted with a later finding in both MEF and HEK293 cells that Gln promoted Rag GTPase-independent lysosomal translocation and activation of mTORC1 [29]. In another study, the addition of bacterial GLS, which hydrolyzes Gln to Glu, to the culture medium or removal of Gln from the medium inhibited mTORC1 and increased LC3-II levels in leukemic cells [53]. Gln withdrawal also increased autophagosome biogenesis in porcine intestinal epithelial cells, with accompanied increase in AMPK but decrease in mTORC1 activities [54].

As a by-product of glutaminolysis, ammonia has also been shown to regulate autophagy, but the emphasis was placed on the induction of autophagy by ammonia [55]. The ability of ammonia to induce autophagy was later shown to be ATG5-dependent but ULK-independent, mimicking that caused by glucose deprivation [42]. Importantly, the ammonia-induced increase in autophagy was not detected until after 4 h of the treatment of the conditioned medium or NH_4_OH [55], suggesting this to be a slow response. Nonetheless, the continued release of ammonia from cells grown in the normal culture medium containing 4 mM Gln likely contributes in part to basal autophagy. On the other hand, ammonia crosses not only the plasma membrane, but organellar membranes as well. This would impact the lysosomal pH. The inhibitory effect of ammonia on lysosomal function has long been known [56,57]. It has also been reported that amino acid withdrawal quickly (within 2 h) resulted in lysosomal acidification, although the exact pH change was not reported [58]. Given that Gln is the main source of ammonia production by the cell, it is not surprising that Gln is the species responsible for the amino acid withdrawal-induced lysosomal pH decrease and the enhanced lysosomal degradation.

Using dextran-conjugated ratiometric pH indicators, we determined the basal lysosomal pH to have a median value around 5.5 and 5.7 in MEF and L929 cells, respectively, kept in normal culture medium at 37 °C. These values are higher than the commonly reported mean luminal pH of lysosomes (pH 4.6–pH 5.2) [3,20,59]. However, upon removal of Gln from the medium, the median lysosomal pH dropped 0.6–1.0 pH unit to values that resemble the commonly reported lysosome pH. Considering that almost all lysosomal pH measurements were carried out in physiological buffers without Gln, and mostly at the room temperature, our results are in excellent agreement with the reported lysosomal pH in the literature. By performing the measurement in the complete medium, including FBS, at 37 °C, we uncovered that for cultured cells, the lysosomal pH values are higher than previously thought. More importantly, the lysosomal pH exhibits a dependence on extracellular Gln concentrations between 0 to 4 mM (Appendix A). Given that the serum Gln concentrations range from 0.2 to 4 mM [25,26], this provides a mechanism to fine-tune lysosomal pH to regulate the degradation capacity of lysosomes based on changes in Gln contents in serum, and hence interstitium.

In time-lapse imaging, the exogenously added Gln to the prior Gln-deprived cells immediately elevated lysosomal pH with no apparent delay, reaching the peak value of ~pH 5.8 within 90 s before declining slightly to a new steady-state level. This response pattern demonstrates a dynamic interplay between V-ATPase, which acidifies the lysosomes, and GLS, which overcomes the acidification by producing ammonia from Gln. That GLS1 and its production of ammonium are ultimately responsible for the Gln-induced lysosomal pH increase was demonstrated by the marked inhibition of this response by either a GLS1 inhibitor or an ammonium scavenger (Figure 5A). These agents suppressed the ability of Gln to overcome lysosomal acidification by either slowing down the production of ammonia or limiting its availability. Consequently, they also accelerated lysosomal degradation in the presence of Gln (Figure 5B,C,F).

### 4.2. Multiple Mechanisms Work in Concert to Regulate Lysosomal pH

Although V-ATPase is responsible for pumping the protons to the lysosome lumen, other mechanisms exist to tune the luminal pH to optimal values to meet the needs of different cellular functions. Ideally, both fast and slow mechanisms should be in place for lysosomal pH regulation. The recent demonstration of the lysosomal proton-activated proton-leak channel (LyPAP or TMEM175) unveils how excess luminal pH drop can be instantly corrected [60]. Our findings, on the other hand, reveal how lysosomal pH can be quickly lowered in response to amino acid shortage. Interestingly, ammonium has been suggested to play a positive role in supporting mTOR activities in multiple cell lines [61]. Therefore, the by-product from glutaminolysis of Gln does not simply slow down degradation, it also plays a positive role in supporting synthetic activities.

Previously, both mTORC1-independent and mTORC1-dependent mechanisms have been shown to influence the lysosomal acidity and thereby lysosomal degradation. The former includes a rapid lysosomal acidification response to amino acid withdrawal that occurs independently of autophagosome formation, as it was unaffected by the knockout of ULK1/2 or ATG5 [58]. The latter appeared during complete serum and amino acid starvation or mTORC1 inactivation by direct mTOR blockers and exhibited a dependence on ATG5/ATG7 [62]. However, in this case, the knockout of ATG5 did not completely eliminate the starvation-induced lysosomal acidification (Supplementary Figure S6 of ref. [62]), implicating that nutrient deprivation exerts an additional effect on lysosomal pH than that triggered by mTOR inactivation. Such an effect, as well as the ULK1/2- and ATP5-independent effect of amino acid withdrawal on lysosomal acidification [58], can be explained by our finding that Gln in the medium produces ammonia as a part of glutaminolysis to dampen the lysosomal pH under normal culture conditions.

The mTORC1-independent lysosomal acidification likely represents an early response to amino acid deprivation since it precedes mTORC1 inactivation and autophagosome formation [58]. The amino acid withdrawal-induced lysosomal acidification was found to be partially prevented by removing glucose or stimulating AMPK via unknown mechanisms [58]. Plausibly, this has to do with the glucose dependence of V-ATPase and its inhibition by AMPK [63,64], as the V-ATPase activity is needed to bring down lysosomal pH when ammonium production ceases as a result of Gln depletion. The later, mTORC1 inactivation-induced, lysosomal acidification, however, could result from increased V-ATPase subunit expression as a part of transcription factor EB (TFEB)-regulated lysosome biogenesis pathway commonly associated with autophagy [65,66], although other unknown mechanisms cannot be excluded. Taken together, both short-term and long-term mechanisms exist for regulation of lysosomal pH. By producing ammonium via glutaminolysis, Gln keeps lysosomal pH at slightly elevated levels, which not only helps support mTOR signaling, but also allows a quick boost of amino acid production through accelerated degradation when these substrates are in short supply.

### 4.3. Glutamine Regulates Lysosomal Degradation and Early Responses of the Cell to Amino Acid Starvation

The elevated lysosomal pH may be advantageous for cell survival under well-fed conditions. Although low luminal pH is better suited for lysosomal digestion, it is not ideal for anabolism because mTORC1 activation is dependent on luminal amino acids [30,31,32,67]. It has been reported that an elevated lysosomal pH facilitates amino acid accumulation inside lysosomal lumen, probably by lowering the activity of lysosomal amino acid transporters [68]. Thus, when nutrients are plentiful and cells thrive on amino acids taken up from extracellular space, the internally generated amino acids through lysosomal degradation mainly serve to support mTORC1 activation. Keeping them in the lysosome with elevated luminal pH would be preferred. However, upon depletion of extracellular amino acids, cells become dependent on internally generated building blocks. Then, lowering the luminal pH accelerates not only the production of amino acids in the lysosome but also their export, which help support mTORC1 activation and protein synthesis, respectively. By producing ammonia to raise lysosomal pH under well-fed conditions and allow it to quickly return to lower values in response to amino acid withdrawal, Gln represents the single most important amino acid species that ensures anabolic activities not to be easily interrupted by a brief disruption of the amino acid supply.

Supporting the above view and consistent with the previous report [69], our data showed that mTORC1 activity did not decrease until after 15 min of amino acid withdrawal (Figure 6C). By contrast, lysosomal degradation increased within 5 min of amino acid withdrawal (Figure 6B). Importantly, the addition of Gln alone prevented the acceleration of degradation and caused more immediate mTORC1 inactivation, supporting the pivotal role of Gln regulation of lysosomal pH in the cellular response to amino acid starvation. It was recently shown that degradation of autophagy receptors represents an immediate autophagic response to amino acid shortage, which occurs independently of mTOR and macroautophagy [17]. The authors have referred this process as endosomal microautophagy. We show here that the removal of Gln alone can accelerate degradation of selective autophagy receptors through decreasing luminal pH. This immediate response to amino acid deprivation likely represents a general effect on accelerating degradation of all proteins destined to the lysosomes, including the endocytosed proteins (Figure 7). The decreases in the steady-state levels of LC3-II and selective autophagy receptors and acceleration of endocytic degradation, thus, reflect a boost in utilizing all available materials to help cells continue with their normal activities.

However, at least for cultured cells examined in the current study, the capacity of this Gln-dependent salvation pathway appears to be limited, as mTORC1 became inactivated after 15 min of amino acid withdrawal, meaning that the readily available reserve of the cell, including the utilization of endocytosed proteins, may only sustain its normal activity for about 15 min. If the amino acid supply does not resume by that time, the cell will initiate macroautophagy, a process involving more drastic changes to conserve energy and resources to support only essential activities for survival. Thus, the Gln depletion-induced increase in lysosomal degradation represents the early phase of the cellular response to amino acid starvation independently of mTORC1 inactivation. This defines the grace window period that the cell can withstand amino acid shortage without having to stop most of its synthetic activities.

Since almost all studies on mTOR and macroautophagy used 50 min or longer amino acid starvation [30,67], and many of them removed both serum and amino acids [62], our current knowledge on starvation-induced changes in lysosomal function was mainly built on the late phase after macroautophagy has initiated. At this stage, a high rate of lysosomal degradation is still needed to match the increased rate of autophagosome synthesis. Most likely, the lysosomal pH remains low because of the low glutaminolysis rate. Additionally, TFEB responds to mTORC1 inactivation to promote lysosome biogenesis, including more V-ATPase production [65,66,70]. This de novo pathway could take hours to become effective [71]. Notably, lysosome function can be upregulated by mTOR inhibitors (PP242 and Torin1) within 3 h in the absence of starvation, indicating that mTORC1 inactivation in the presence of Gln is able to enhance lysosomal degradation [62]. Therefore, multiple mechanisms are involved to boost lysosome function at different stages of nutrient deprivation to help cells adapt the changing environment for survival.

In summary, our data reveal a previous unknown effect of Gln on lysosome function that occurs through ammonium produced as a by-product of glutaminolysis. Under well-fed conditions, ammonia diffuses to the lysosome to elevate lysosomal pH, which not only slows down degradation but also allows amino acid accumulation in the lysosomal lumen to facilitate mTORC1 activation. Upon depletion of amino acids, the loss of Gln abolishes ammonium production, leading to a rapid decrease in lysosomal pH, which augments lysosomal degradation to generate amino acids from internal reserves and endocytosed proteins. This Gln-dependent pathway is particularly important for cells to continue their normal synthetic activities without an interruption in times when amino acid supply is briefly disrupted.

## Figures and Tables

**Figure 1 cells-12-00080-f001:**
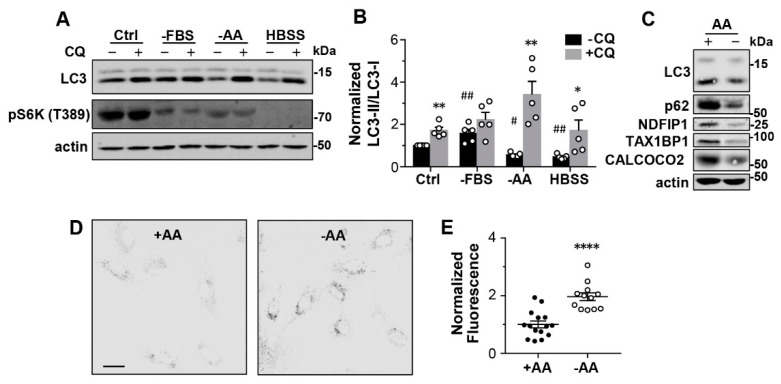
Amino acid starvation accelerates lysosomal degradation. (**A**) Representative Western blot analysis of LC3, phospho-S6K (T389) (pS6K), and actin of MEF cells cultured in normal medium (Ctrl) or subject to 1-h serum starvation (−FBS), amino acid starvation (−AA), or complete starvation (HBSS) in the absence (−) or presence (+) of chloroquine (CQ, 100 µM). (**B**) Statistics of LC3-II/LC3-I ratios normalized to that of untreated Ctrl for conditions shown in (**A**) from *n* = 5 experiments. * *p* < 0.05, ** *p* < 0.01 vs. corresponding +CQ by Grouped analyses—multiple *t* tests; ^#^ *p* < 0.01, ^##^ *p* < 0.01 vs. Ctrl−CQ, by one-way ANOVA with Dunnett’s multiple comparisons test. (**C**) Western blot analysis of LC3, p62, NDFIP1, TAX1BP1, CALCOCO2, and actin of primary human skin fibroblasts, BJ cells, cultured in normal medium or subject to 1-h amino acid starvation. (**D**) Representative images of DQ-BSA loaded MEF cells cultured in normal medium (+AA) or subject to 1-h amino acid starvation (−AA). Scale bar, 20 µm. (**E**) Statistics of fluorescence intensities of individual cells normalized to the mean of the +AA controls. **** *p* < 0.0001 by *t* test.

**Figure 2 cells-12-00080-f002:**
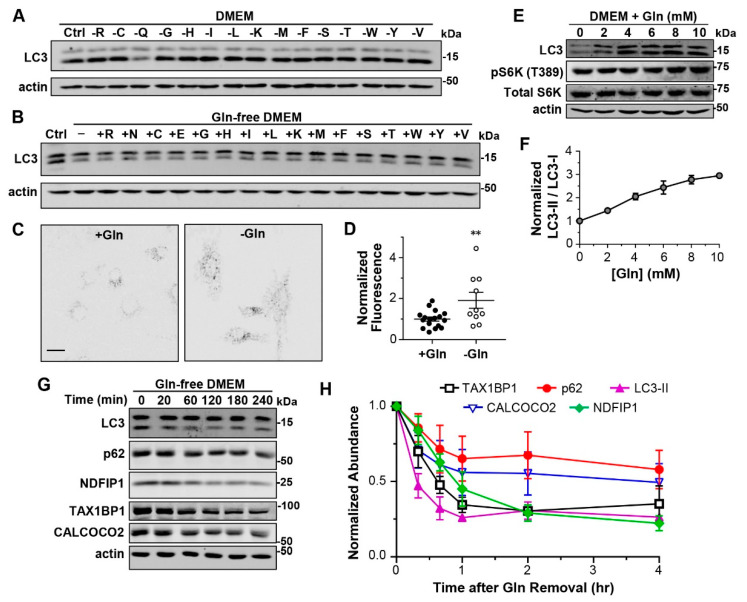
Glutamine is responsible for accelerating lysosomal degradation upon amino acid withdrawal. (**A**) Representative Western blot analysis of LC3 and actin of MEF cells subject to 1-h treatment in DMEM media omitting individual amino acids as indicated. Single-letter amino acid code is used. (**B**) Representative Western blot analysis of LC3 and actin of MEF cells that were treated for 1 h with Gln-free DMEM supplemented with 4 mM of the indicated amino acids. (**C**) Representative images of DQ-BSA-loaded MEF cells cultured in normal medium (+Gln) or subject to 1-h Gln starvation (−Gln). Scale bar, 20 µm. (**D**) Statistics of fluorescence intensities of individual cells normalized to the mean of the +Gln controls. ** *p* < 0.01, by *t* test. (**E**) Representative Western blot analysis of LC3, pS6K (T389), total S6K, and actin in MEF cells treated for 1 h with DMEM containing indicated concentrations of Gln. (**F**) Concentration response curve of LC3-II/LC3-I ratios to Gln in the culture media, determined from n = 3 experiments as shown in (**E**). (**G**) Western blot analysis of LC3, p62, NDFIP1, TAX1BP1, CALCOCO2, and actin in BJ cells exposed to the Gln-free medium for different time periods as indicated. (**H**) Statistics of LC3-II, p62, NDFIP1, TAX1BP1, and CALCOCO2 normalized to untreated BJ cells during 4-h time course of Gln starvation, from *n* = 3 experiments.

**Figure 3 cells-12-00080-f003:**
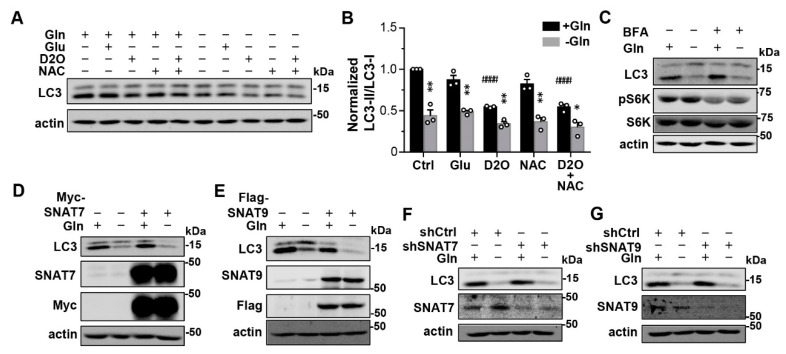
Glutamine regulates lysosomal degradation through a mechanism unrelated to oxidation, Gln-driven energy metabolism, or lysosomal amino acid sensing. (**A**) Representative Western blot analysis of LC3 and actin of MEF cells treated with Glu (4 mM), DM-2-OG (D2O, 6 mM), and/or NAC (6 mM), as indicated, in the presence or absence of Gln for 1 h. (**B**) Statistics of LC3-II/LC3-I ratios normalized to that of untreated Ctrl for conditions shown in (**A**) from *n* = 3 experiments. Note, none of the treatments affected the Gln withdrawal effect; D2O also decreased LC3-II in the control medium. * *p* < 0.05, ** *p* < 0.01 vs. corresponding +Gln by Grouped analyses—multiple *t* tests; ^####^ *p* < 0.0001 vs. Ctrl + Gln, by one-way ANOVA with Dunnett’s multiple comparisons test. (**C**) Western blot analysis of LC3, pS6K (T389), total S6K, and actin of MEF cells untreated or treated with BFA (10 µM) in the presence and absence of Gln for 1 h. (**D**,**E**) Western blot analysis of LC3, SNAT7, Myc tag, and actin of MEF cells that overexpressed Myc-SNAT7 (**D**) and of LC3, SNAT9, FLAG tag, and actin of MEF cells that overexpressed Flag-SNAT9 (**E**). Ectopic expression was achieved via lentiviral transduction. Cells were kept in the normal culture medium or subject to Gln withdrawal for 1 h. (**F**,**G**) Western blot analysis of LC3 and actin of MEF cells that received control shRNA (shCtrl), or shRNA against SNAT7 (shSNAT7, (**F**)) or SNAT9 (shSNAT9, (**G**)) via lentiviral transduction. Cells were kept in the normal culture medium or subject to Gln withdrawal for 1 h. Knockdown efficiency was evaluated by Western blotting with anti-SNAT7 (**F**) and anti-SNAT9 (**G**) antibodies.

**Figure 4 cells-12-00080-f004:**
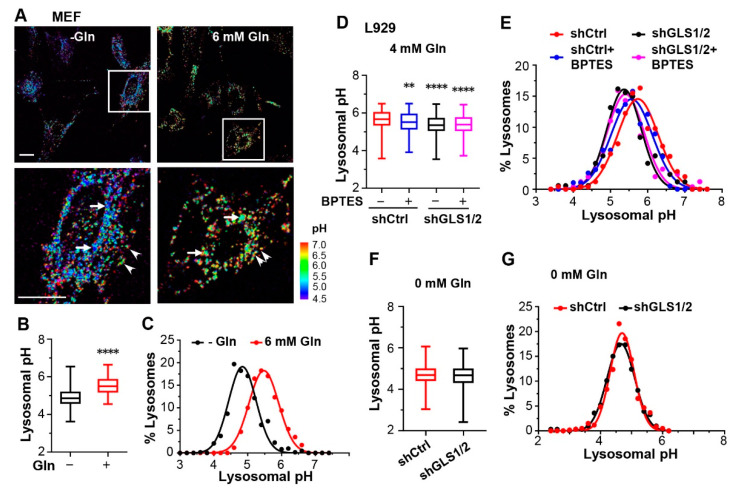
Glutamine alkalizes lysosomes in a glutaminase-dependent manner. (**A**) Representative pseudocolor images of lysosomal pH derived from ratiometric confocal micrographs of MEF cells loaded with Oregon Green 488 dextran (OG-488) and cultured in media with 0 (−Gln) and 6 mM Gln. Boxed areas are enlarged below to show individual lysosomes and lysosome clusters. Note the lower pH for lysosomes near the nuclei (arrows) than those in the cell peripheral (arrowheads) in both 0 and 6 mM Gln. Scale bars, 20 µm. (**B**) Statistics of lysosomal pH values from MEF cells measured as in (**A**) for 609 and 170 individual lysosomes or lysosome clusters in 0 and 6 mM Gln, respectively, from 30–100 cells pooled from *n* = 3 experiments. For all box and whisker plots, middle lines represent median values; boxes and whiskers represent 25 to 75 percentiles and min to max values, respectively. **** *p* < 0.0001 by *t* test. (**C**) Gaussian distribution analysis of the same data from (**B**). (**D**,**E**) Steady-state lysosomal pH of L929 cells transfected with either shCtrl or shRNAs against GLS1 and GLS2 (shGLS1/2). Cells were loaded with FITC-dextran and cultured in medium with 4 mM Gln. BPTES (20 µM) was added as indicated after the chase of the dextran dye and allowed to incubate with the cells for 2 h before imaging. Shown are median values (**D**) and Gaussian distribution analyses (**E**) of pH values of 440, 189, 159, and 229 individual lysosomes or lysosome clusters for shCtrl-BPTES, shCtrl+BPTES, shGLS1/2-BPTES, shGLS1/2+BPTES, respectively, from 30–100 cells pooled from *n* = 3 experiments. ** *p* < 0.01, **** *p* < 0.0001, by one-way ANOVA with Tukey’s multiple comparisons test. (**F**,**G**) Steady-state lysosomal pH measured by FITC-dextran of L929 cells transfected with either shCtrl or shGLS1/2. Cells were cultured in the Gln-free medium for 1 h. Shown are median values (**F**) and Gaussian distribution analyses (**G**) of pH values of 598 and 306 individual lysosomes or lysosome clusters for shCtrl and shGLS1/2, respectively, from 30–100 cells pooled from *n* = 3 experiments.

**Figure 5 cells-12-00080-f005:**
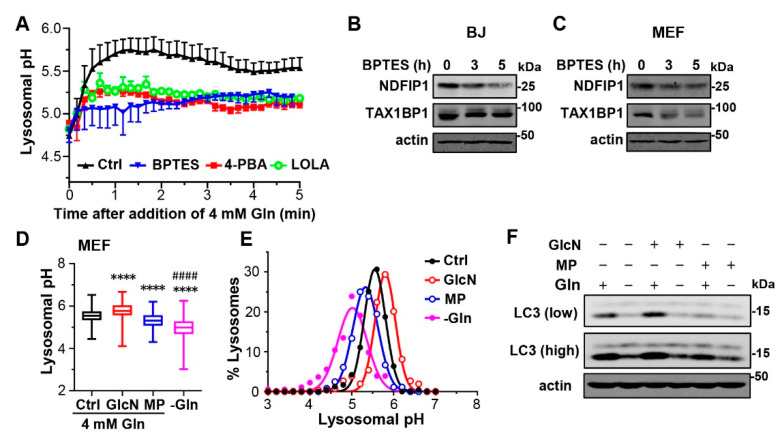
GLS1 inhibitor and ammonium scavengers attenuate glutamine effects on lysosomal pH and degradation. (**A**) Kinetics of lysosomal pH changes of L929 cells in response to the addition of Gln. FITC-dextran-loaded cells were cultured in Gln-free medium for 1 h before 2 mM Gln was added back. BPTES (20 µM), 4-PBA (10 mM), or LOLA (10 mM) was added after the chase of the dextran dye and allowed to incubate with the cells for 2 h (including the 1-h Gln starvation). Confocal ratiometric images were acquired at 10-s intervals immediately after the application of 2 mM Gln and continued for 5 min in an imaging chamber with the cells maintained live at 37 °C. Data points are means ± SEM of *n* = 3 experiments, each representing average lysosomal pH values of 30–50 cells. (**B**,**C**) Western blot analysis of NDFIP1, TAX1BP1, and actin of BJ (**B**) and MEF (**C**) cells cultured in normal medium without or with BPTES (20 µM) for 3 and 5 h as indicated. (**D**) Statistics of lysosomal pH of MEF cells maintained in normal culture medium supplemented with either glucosamine (GlcN, 10 mM) or methyl pyruvate (MP, 20 mM) as indicated, or in the Gln-free medium, based on quantification of 235 (Ctrl), 259 (GlcN), 360 (MP), 356 (−Gln) individual lysosomes or lysosome clusters from 30–100 cells pooled from *n* = 3 experiments. **** *p* < 0.0001 vs. Ctrl, ^####^ *p* < 0.0001 vs. either GlcN or MP, by one-way ANOVA with Tukey’s multiple comparisons test. (**E**) Gaussian distribution analysis of the same data from (**D**). (**F**) Western blot analysis of LC3 and actin of MEF cells cultured in normal medium supplemented with GlcN (10 mM) or MP (20 mM) as indicated in the presence or absence of Gln (4 mM). Low and high mean short and long exposures, respectively.

**Figure 6 cells-12-00080-f006:**
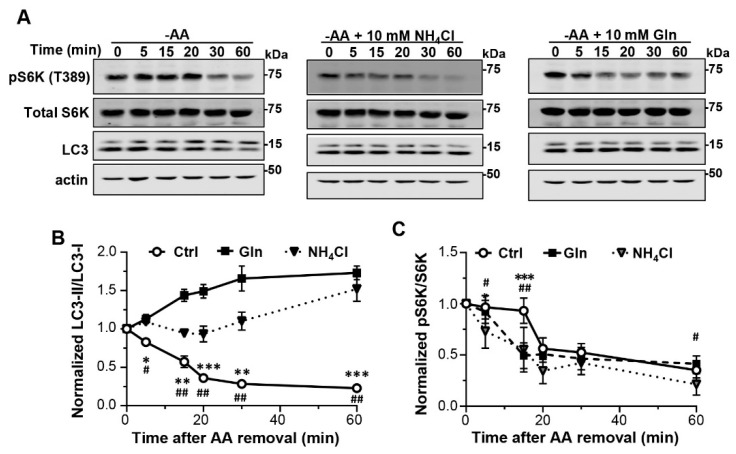
Inhibition of lysosome degradation accelerates mTORC1 inactivation during early phase of amino acid starvation. (**A**) Representative Western blot analysis of LC3, pS6K (T389), total S6K, and actin of MEF cells exposed to amino acid-free medium (−AA) for different time periods as indicated. NH_4_Cl (10 mM) and Gln (10 mM) were included in the amino acid-free medium as shown. (**B**,**C**) Statistics of LC3-II/LC3-I (**B**) and pS6K/total S6K (**C**) ratios normalized to that of unstarved cells for conditions shown in (**A**). Data points are means ± SEM from *n* = 3 experiments for (**B**) and *n* = 6 experiments for (**C**) except for the 20 min data points, which had *n* = 3. * *p* < 0.05, ** *p* < 0.01, *** *p* < 0.001, Ctrl vs. Gln; ^#^ *p* < 0.05, ^##^ *p* < 0.01, Ctrl vs. NH_4_Cl, at each of the time points by Grouped analyses—multiple *t* tests. For (**B**), *p* < 0.05 between time points; *p* < 0.0001 between treatment conditions; *p* < 0.0001 for interaction between time points and treatment conditions, by two-way ANOVA. For (**C**), *p* < 0.0001 between time points; *p* < 0.0001 between treatment conditions; *p* < 0.0001 for interaction between time points and treatment conditions, by two-way ANOVA.

**Figure 7 cells-12-00080-f007:**
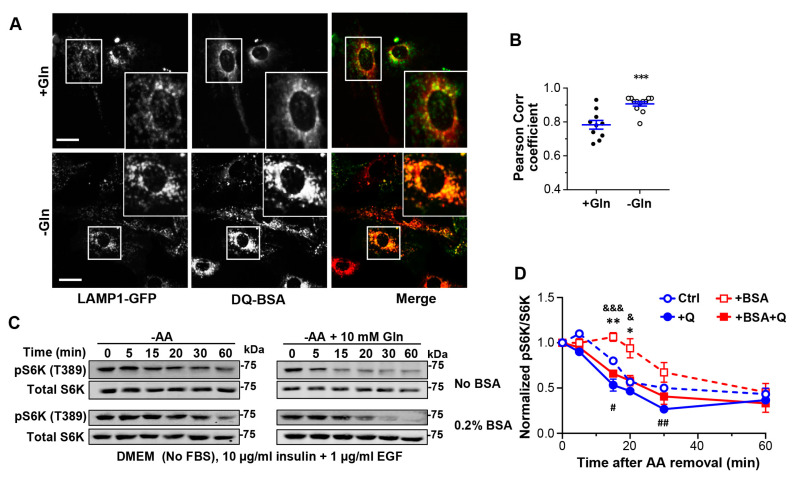
Glutamine deprivation enhances protein endocytosis to sustain mTORC1 activity. (**A**) Representative images of DQ-BSA (*red*) taken up by MEF cells expressing LAMP1-GFP (*green*). Cells were exposed to DQ-BSA in medium containing 4 mM Gln (+Gln) or no Gln (−Gln) for 1 h. (**B**) Statistical analysis of the Pearson correlation coefficient between DQ-BSA and LAMP1-GFP for (**A**). Data points were from 3–4 coverslips with 3 areas of each coverslip, *** *p* < 0.001 by *t* test. (**C**) Representative Western blot analysis of pS6K (T389) and total S6K of MEF cells cultured in serum-free media supplemented with 10 µg/mL insulin and 1 µg/mL EGF without (*upper two rows*) or with 0.2% BSA (*lower two rows*). The cells were deprived of amino acids (−AA) for different time periods as indicated. Gln (10 mM) was included during the amino acid starvation as shown. (**D**) Statistics of pS6K/total S6K ratios normalized to that of unstarved cells (Time 0) for conditions shown in (**C**). Data points are means ± SEM from *n* = 3–5 experiments [3 for Ctrl (−AA) and +Q, 4 for +BSA, 5 for +BSA + Q]. * *p* < 0.05, ** *p* < 0.01 Ctrl vs. +BSA; ^#^ *p* < 0.05, ^##^ *p* < 0.01 Ctrl vs. +Q; ^&^ *p* < 0.05, ^&&&^ *p* < 0.001 +BSA vs. +BSA + Q, at each of the time points by Grouped analyses—multiple *t* tests. *p* < 0.0001 between time points; *p* < 0.0001 between treatment conditions; *p* < 0.01 for interaction between time points and treatment conditions, by two-way ANOVA.

## Data Availability

Data supporting the reported results are available from the corresponding author upon request.

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
