# Peer review of "Glutamine Produces Ammonium to Tune Lysosomal pH and Regulate Lysosomal Function"

_cells, 2022, doi:10.3390/cells12010080_

Round 1

Reviewer 1 Report

In this study, Xiong et al report that extracellular glutamine regulates the lipid-conjugated form of LC3 (LC3-II), an ATG8 family autophagy protein, and autophagy cargo receptors through a modulation of lysosomal pH and lysosomal proteolysis by glutaminolysis-derived ammonium. The study thus adds another layer of lysosome and autophagy regulation by intracellular amino acids which is independent of, and faster than, mTORC1 signaling. In a final twist the authors suggest that this mechanism could help maintain mTORC1 activity during brief amino acid shortages by upregulating (auto)lysosomal proteolysis and amino acid supply by basal autophagy or, in some cell types, by protein endocytosis.

Overall this is an interesting study which unveils a novel effect of intracellular amino acids on lysosomes and clarifies discrepancies about the effects of glutamine reported in the literature. However, although the narrative is clear and experiments are well described, the manuscript is difficult to follow because the evidence is not presented in a straightforward manner and some key pieces of evidence are missing.

Major points:

1.     Showing LC3-II to LC3-I ratio graphs rather than LC3-II levels to draw conclusions in the main figures is quite confusing because these ratios do not directly reflect the process reported by the authors: the degradation of LC3. As this reviewer understood, their working model posits that the LC3-II form is selectively degraded upon glutamine deprivation through a process reported by Mejlvang et al 2018 (ref. 17) for the downregulation autophagy cargo receptors by amino acid starvation. LC3-II levels provide more direct evidence. These graphs should be moved from supplementary to main figures. It would also be useful to document (in supplements) that LC3-I levels are not significantly affected under the tested conditions.

2.       The link with the downregulation of autophagy receptors is documented at the beginning of the study (Figs 1C and 2G,H). The authors show that this process also selectively depends on glutamine. However, they do not pursue this research line. Whether this fast, mTORC1-independent downregulation of cargo receptors also proceeds through glutamine catabolism and ammonium production has not been tested. This question should be addressed for at least one ‘well-responding’ cargo receptor, for instance TAX1BP1.

3.       Mejlvang et al reported that cargo receptor downregulation by amino acid stravation proceeds through microautophagy in late endosomes. Does microautophagy act upstream of the glutamine-dependent selective degradation of LC3-II? As conjugation of ATG8 to single membranes (CASM) in endolysosomes requires ATG8 lipidation to PE or PS, such a delivery pathway could help explain why LC3-II is selectively affected.

4.       Treatment with a membrane permeable analog of a-ketoglutarate substantially impairs the down-regulation of LC3-II by glutamine deprivation (Fig. S3A; the effect is also reflected in Fig. 3B). This should be commented.

5.       The GLS knock-down experiments (Figs 4E,F) are poorly convincing, possibly because residual enzyme activity mitigates the dependence on glutamine at steady state. Time-lapse lysosomal pH recording, which show a fast, robust response to glutamine (Fig. 4G), would help address this pitfall. shGLS1/2-treated cells should show a slower response to glutamine. This approach could provide stronger molecular evidence for the involvement of GLS 1 and/or 2.

6.       The concentration used (10 mM) for exogenous applications of NH4+ is quite high. Have the authors tested lower concentrations?

7.       The evidence provided for the upregulation of protein endocytosis by glutamine deprivation (Fig. 6) is not convincing because the use of DQ-BSA probe makes it difficult to discriminate between enhanced endocytosis and enhanced proteolysis effects.

Minor:

1.       The pH differences between central and peripheral lysosomes (p. 10, lines 443-445) are not visible in Fig. 4A. Showing an enlarged cell for each condition would help.

2.       The reason stated for preferring FITC over Oregon Green (p. 10, line 460) is not convincing. In fact, the pH calibration curves in Fig. S4B suggest OG-dextran provides a better match.

Author Response

We thank reviewer #1 for the general support of the overall story. We have significantly revised the manuscript to improve the clarity. This includes splitting the original Fig. 4 into two figures and adding additional data on autophagy receptor degradation by the inhibition of GLS1 (new Fig. 5B, C). Please see the attachment for detailed response.

Reviewer 2 Report

Glutamine is one of the most abundant amino acids in the cell. It has been implicated in mTOR-autophagy regulation, but with controversy. In this study, the authors revealed a previous unknown effect of glutamine on lysosome function that occurs through ammonium produced as a by-product of glutaminolysis. Under fed conditions, glutamine alkalizes lysosomes by producing ammonium through glutaminases, which not only slows down degradation but also allows amino acid accumulation in the lysosomal lumen to facilitate mTORC1 activation. Upon depletion of amino acids (the early stage of amino acid starvation), the loss of glutamine abolishes ammonium production, leading to a rapid decrease in lysosomal pH. This further augments lysosomal degradation to generate amino acids from internal reserves and endocytosed proteins, sustaining mTORC1 activity. The study is very interesting and significant. Their conclusions were strongly supported by solid evidence. The paper was well written with comprehensive discussion.

Some suggestions for the authors to consider:

Figure 1A-B:

The data clearly demonstrated that AA starvation promotes lysosomal degradation. According to the marked decrease in the level of phospho-p70S6K (pS6K), the author claimed that in the starved cells macroautophagy was triggered under all starvation conditions. However, no difference of the LC3II levels between Ctrl and HBSS after CQ treatment. This could be discussed.  

Higher level of LC3 in AA starvation than that in HBSS starvation. This is likely attributed to something in DMEM that can promote autophagosome biogenesis. This also could be discussed.

Figure 2E:

As in Fig. 1, measurement of LC3II levels upon CQ upon glutamine starvation is helpful, but not necessary.  

Author Response

Reviewer #2

Comments and Suggestions for Authors

Glutamine is one of the most abundant amino acids in the cell. It has been implicated in mTOR-autophagy regulation, but with controversy. In this study, the authors revealed a previous unknown effect of glutamine on lysosome function that occurs through ammonium produced as a by-product of glutaminolysis. Under fed conditions, glutamine alkalizes lysosomes by producing ammonium through glutaminases, which not only slows down degradation but also allows amino acid accumulation in the lysosomal lumen to facilitate mTORC1 activation. Upon depletion of amino acids (the early stage of amino acid starvation), the loss of glutamine abolishes ammonium production, leading to a rapid decrease in lysosomal pH. This further augments lysosomal degradation to generate amino acids from internal reserves and endocytosed proteins, sustaining mTORC1 activity. The study is very interesting and significant. Their conclusions were strongly supported by solid evidence. The paper was well written with comprehensive discussion.

We thank reviewer #2 for the positive comments and the encouraging support.

Some suggestions for the authors to consider:

Figure 1A-B:

The data clearly demonstrated that AA starvation promotes lysosomal degradation. According to the marked decrease in the level of phospho-p70S6K (pS6K), the author claimed that in the starved cells macroautophagy was triggered under all starvation conditions. However, no difference of the LC3II levels between Ctrl and HBSS after CQ treatment. This could be discussed. 

Higher level of LC3 in AA starvation than that in HBSS starvation. This is likely attributed to something in DMEM that can promote autophagosome biogenesis. This also could be discussed.

We agree with reviewer #2 that these are interesting observations. However, they are not easy to interpret. The main differences in AA-free DMEM medium (with serum) and HBSS are the presence of FBS and higher glucose concentration in the former. There are also more antioxidants and lipids in the culture medium than in HBSS. Therefore, the AA-free DMEM medium should support cell survival better than HBSS. We could speculate that perhaps in HBSS, LC3-II synthesis is increased very slowly due to, for example, a lack of sufficient fatty acids for lipidation. This might be why in most studies using HBSS, a long starvation time, typically 4 to 12 hours, is required to reveal autophagosome synthesis. Since complete nutrient deprivation is not the focus of the current study, an extensive discussion on HBSS-induced effects will detract the main message of our report.

Figure 2E:

As in Fig. 1, measurement of LC3II levels upon CQ upon glutamine starvation is helpful, but not necessary. 

Agreed. Having such data would probably reveal if there is glutamine dependence for autophagosome synthesis. We do not expect this to be the case given the lack of effect of glutamine on pS6K levels, i.e., macroautophagy. Since the question is related to autophagosome synthesis rather than degradation and the journal only gave us 10 days to submit the revision, we will have to defer this to a future study.

Reviewer 3 Report

Usually, cells degrade proteins either through the Ubiquitin Proteasome System or through autophagy. To many extents, both rely on the Ubiquitin signal, though some forms of autophagy are Ubiquitin independent.  Differently from UPS-mediated protein degradation, the autophagy degradation system derives its degradative abilities from the lysosome by means of the hydrolases residing in its lumen. Notably, these enzymes display optimal pH in the acidic range. Thus, lysosomal pH changes negatively impact the hydrolases' efficacy and eventually the lysosome function by itself. Xiong J and colleagues in their manuscript titled "Glutamine tunes lysosomal pH to regulate lysosomal function through production of ammonium", prompted by previous evidence that linked the aminoacid deprivation/starvation to the autophagic process, provide evidence that the aminoacid glutamine indirectly, by means of the glutaminases-dependent production of the ammonium, regulates the lysosomal activity. Overall, I appreciated the manuscript from the rationale to the approaches used to provide the experimental evidence that is quite fully supporting their conclusions. The manuscript is well-written and articulated with sufficient scientific soundness. Prior to publication, there are just a couple of issues that can be improved, and they concern the artwork. Figures 1, 2, 4, and 6 all of them contain a panel with pictures of cells. These panels need to be slightly magnified because as such when printed it is quite hard to properly appreciate the contents being tiny and small intracellular vesicles.

Author Response

Reviewer #3

Comments and Suggestions for Authors

Usually, cells degrade proteins either through the Ubiquitin Proteasome System or through autophagy. To many extents, both rely on the Ubiquitin signal, though some forms of autophagy are Ubiquitin independent.  Differently from UPS-mediated protein degradation, the autophagy degradation system derives its degradative abilities from the lysosome by means of the hydrolases residing in its lumen. Notably, these enzymes display optimal pH in the acidic range. Thus, lysosomal pH changes negatively impact the hydrolases' efficacy and eventually the lysosome function by itself. Xiong J and colleagues in their manuscript titled "Glutamine tunes lysosomal pH to regulate lysosomal function through production of ammonium", prompted by previous evidence that linked the aminoacid deprivation/starvation to the autophagic process, provide evidence that the aminoacid glutamine indirectly, by means of the glutaminases-dependent production of the ammonium, regulates the lysosomal activity. Overall, I appreciated the manuscript from the rationale to the approaches used to provide the experimental evidence that is quite fully supporting their conclusions. The manuscript is well-written and articulated with sufficient scientific soundness. Prior to publication, there are just a couple of issues that can be improved, and they concern the artwork. Figures 1, 2, 4, and 6 all of them contain a panel with pictures of cells. These panels need to be slightly magnified because as such when printed it is quite hard to properly appreciate the contents being tiny and small intracellular vesicles.

We thank reviewer #3 for the positive comments and the encouraging support. We have revised the figures according to the suggestions. Images in Figs 1, 2, 4, 6 (now 7) are enlarged. For images in Fig. 4A, we also enlarged single cells with higher magnifications. Fig. 1D and Fig. 2C are shown again in supplementary Figs. 1 and 2 with black background and red color to reflect the fluorescence of DQ Red BSA, with single cells enlarged for better viewing of intracellular vesicles.

Reviewer 4 Report

The manuscript by Xiong et al is interesting because it deals with a hot topic in science that is the role of glutamine in lysosomal biology with a particular reference to ammonium that is a byproduct of glutamine metabolism often underconsidered in literature. However a few concerns arose which are listed below:

1) I would suggest changing the title that, in this version, is not easily readable.

2) Introduction: the concepts written in lines 60-62 overlap those written in lines 68-69. I would suggest avoid such a close repetition.

3) Results: lines 262-263: please comment on the reported finding.

4) Results: figure 1B is not cited in the text; what is the ratio LC3II/LC3I? Figure 1D-E, how do the authors conclude that the degradation already existed in lysosome before aa starvation? May be a further control should be added to the experiment.

5) Fig 2. Are the described effects also evident at mRNA level?

6) Lines 423-426: how did authors prove that snat7 and snat9 with tags reached properly the lysosomal  membrane? An IF with lysosomal marker should be perfomed to validate the system. Moreover, the WB for SNAT7 is too saturated to appreciate a difference, is any, between the tested conditions.

7) To support mTORC activation another downstream of mTORC should be also measured to demonstrate the direct activation of the kinase.

8) Discussion: line 689: SL7A5 is not a glutamine transporter. Please rephrase the sentence indicating other transporters such as SLC1A5, SLC38A2 or others

9) Discussion is in general too much repetitive of results. It needs to be shortened and improved.

10) In spite of the quite convincing results on protein expression and pH measure, the interpretation of ammonia action results very diffucult. Authors must at least discuss how ammonia produced in mitochondria could selectively cause alkalinization of lysosomes. Indeed, CPS1 (Carbamyl phosphste sinthetase 1) may be present in mitochondria of the used cells and remove all produced ammonia. Authors should test CPS1 presence. This is not trivial, since recent findings highlight that cancer cells can express urea cycle enzymes. Alternatively, ammonia should escape mitochondria as NH3 by diffusion and then it can be neutralized in the cytosol by cell buffer systems. Thus, it is difficult to explain how ammmonia could reach lysosome lumen as NH3 for being converted there into NH4+ for alkalynization purpose. Explanation fo such a flux is mandatory for a correct interpretation of the results.

Round 2

Reviewer 1 Report

The authors satisfactorily addressed the points raised, considering the very short revision delay required by this journal.

Author Response

We thank reviewer #1 for his/her support.

Reviewer 4 Report

Authors improved the manuscript. The responses are more or less convincing. However, point 10 of my comments should be more adequately addressed. Therefore:

- the issue should be better commented in discussion explaining the effect of ammonia also in the light of the author response to point 10.

- some comments for excluding mitochondrial action as scavengers of ammonia should be added (see possible function of CPS1 reported in my point 10).

- Ammonia has a pKa higher than 9. This means that at neutral pH it is mostly (about 99%) and immediately protonated to NH4+. Thus, authors have to explain (or just hypothesize) more convincingly than by the acidic pH of lysosomes, why the effect might be more relevant in lysosomes. Refs 56/57 cited by authors do not provide enough (if any) evidences that ammonia can reach lysosomes and modify their metabolism.

Author Response

We thank reviewer #4 for his/her reading of the revised manuscript.

We understand that CPS1 catalyzes synthesis of carbamoyl phosphate from ammonia and bicarbonate in liver and intestinal epithelium, which then enters the urea cycle. However, we do not know how prominent this reaction is in cells not derived from liver or epithelium. Neither do we have any knowledge on how the rate of ammonia consumption by this enzyme compares with ammonia diffusion out of mitochondria. It would be inappropriate for us to comment on something that we do not have sufficient knowledge of.

The reviewer is correct that about 99% ammonium is present in the form of NH4+ at neutral pH. In fact, if we assume a Hill coefficient of 1 (with pKa of 9), NH4+ would account for 98.44% of total ammonium at pH 7.2. However, due to the fact that NH3 is a gas and is freely diffusible across the entire cell including all lipid bilayers, this reaction cannot be kept at equilibrium in any cellular compartment or even the entire cell, as NH3 will be continuously lost. At pH 5.6 and 4.6, the lysosomal pH revealed in the current study in the absence and presence of glutamine, respectively, NH4+ accounts for 99.960% and 99.996%. Given that at equilibrium NH3 accounts for 1.56%, 0.04% and 0.004% of the total ammonium at pH 7.2, 5.6, and 4.6, at the continued disequilibrium more NH4+ will be converted to NH3 at pH 7.2 than at pH 5.6 and 4.6. This will result in faster loss of ammonium at neutral that at acidic pH. This is why ammonia has a more prominent effect on lysosomes than other cellular compartments.

Like chloroquine and bafilomycin A1, NH4Cl is commonly used to inhibit lysosome function. The two references cited represent some very early studies that used NH4Cl to study lysosome function.